# Mapping evidence on the factors contributing to long waiting times and interventions to reduce waiting times within primary health care facilities in South Africa: A scoping review

Ugochinyere I. Nwagbara[1,2]*, Khumbulani W. Hlongwana[1,3], Sylvester C. Chima[2]

1 Discipline of Public Health Medicine, School of Nursing and Public Health, University of KwaZulu-Natal, Durban, South Africa, 2 Programme of Bio & Research Ethics and Medical Law, Nelson R Mandela School of Medicine & School of Nursing and Public Health, College of Health Sciences, University of KwaZulu-Natal, Durban, South Africa, 3 Cancer & Infectious Diseases Epidemiology Research Unit (CIDERU), College of Health Sciences, University of KwaZulu-Natal, Durban, South Africa

* 216045259@stu.ukzn.ac.za, ugochinyereijeoma@gmail.com

**Data Availability Statement:** All relevant data are within the paper and its Supporting information files.

## Abstract

### Background

Globally, reduction of patient waiting time has been identified as one of the major characteristics of a functional health system. In South Africa, 83% of the general population visiting primary healthcare (PHC) facilities must contend with long waiting times, overcrowding, staff shortages, poor quality of care, an ineffective appointment booking system, and a lack of medication. These experiences may, in turn, affect how patients view service quality.

### Methods

This scoping review was guided by Arksey and O'Malley methodological framework. The primary literature search of peer-reviewed and review articles was achieved through PubMed/MEDLINE, Google Scholar, Science Direct, and World Health Organization (WHO) library databases, using waiting times, outpatient departments, factors, interventions, and primary healthcare facilities as keywords. Two independent reviewers screened abstracts and full articles, using the set inclusion and exclusion criteria. We used NVIVO® version 10 software to facilitate thematic analysis of the results from included studies.

### Results

From the initial 250 records screened, nine studies were eligible for inclusion in this scoping review. Seven papers identified the factors contributing to waiting time, and five papers mentioned effective interventions implemented to reduce waiting times within PHC facilities. Our analysis produced three (patient factors, staff factors, and administrative systems) and two (manual-based waiting time reduction systems and electronic-based waiting time reduction systems) main themes pertaining to factors contributing to long waiting times and interventions to reduce waiting times, respectively.

**Funding:** The first author (UIN) received a Tuition Remission for her Ph.D. studies from UKZN and partial scholarship funding from the College of Health Sciences (CHS), UKZN, to support her PhD studies. The funders had no role in study design, data collection and analysis, decision to publish, or preparation of the manuscript.

**Competing interests:** The authors have declared that no competing interests exist.

**Abbreviations:** CHC, Community Health Centre; CTS, Cape Triage Score; LMICs, Low-and-Middle-Income Countries; MESH, Medical Subject Headings; MMAT, Mixed Method Appraisal Tool; PCC, Population Content Context; PHC, Primary Health Care; PRISMA-ScR, Preferred Reporting Items for Systematic Reviews and Meta-Analyses Extension for Scoping Reviews; SCREEN, Sick Children Require Emergency Evaluation Now; WHO, World Health Organization.

## Conclusion

Our results revealed that the patients, staff, and administrative systems all contribute to long waiting times within the PHC facilities. Patient waiting times recorded a wider and more evenly spread patient arrival pattern after the identified interventions in our study were implemented. There is a need to constantly strategize on measures such as implementing the use of an electronic appointment scheduling system and database, improving staff training on efficient patient flow management, and regularly assessing and optimizing administrative processes. By continuously monitoring and adapting these strategies, PHC facility managers can create a more efficient and patient-centered healthcare experience.

## Introduction

Globally, successful reduction of patient waiting times has been identified as one of the major characteristics of a functional health system [1]. South Africa's public health sector is grappling with long waiting times in primary health care (PHC) facilities, which undermine the functionality of health systems. Consequently, these long waiting times culminate into a range of issues, such as patients' dissatisfaction with care, interruption of hospital work patterns, impaired access to care and increased medical errors, and patients having to skip appointments, resulting in them moving from one clinic to the other in search of shorter waiting times [2–6]. Patient waiting time is defined as the amount of time spent by patients seeking care at healthcare units before being attended to for consultation and treatment [7–9].

In South Africa, 83% of the general population visiting the PHC facilities have to contend with long waiting times, attributable to overcrowding, staff shortages, poor care quality, an ineffective appointment booking system, a lack of medication, inadequate infrastructure, a lack of appropriate medical equipment, and poor data management [10, 11]. Reducing long waiting times and overcrowding is crucial for preventing the spread of infections [12]. It has been reported that staff shortages caused by attending training or meetings during working hours also contribute to patients' extended waiting times in PHC facilities [13].

Given the universal nature of long waiting times, the United States Institute of Medicine (IOM) recommends that at least 90% of patients should be seen within 30 minutes of their scheduled appointment time [9, 14]. However, this is not the case in most low-and-middle-income countries (LMICs), and South Africa is no exception as several studies have shown that patients spend between 2–4 hours in outpatient departments before seeing a doctor [15, 16]. Long waiting times negatively affect the overall quality of care, patient safety and satisfaction, and clinic staff morale, and compromise the doctor-patient relationship [8]. In some instances, children brought to PHC centers by carers wait nearly the whole day, only to be turned away at closing time without being seen by any healthcare worker (HCW) [17]. The South African National Department of Health (NDoH) has identified waiting time as one of the six priority areas for improving quality of care [18].

Globally, studies have demonstrated a strong inverse relationship between patient satisfaction and waiting times [19–23]. Patients often spend more time waiting than consulting with healthcare providers [3].

A multidisciplinary team approach remains among the key approaches to rekindle hope that reduced wait times, improved health outcomes, and enhanced patient satisfaction, can be achieved [24]. It is, however, evident that long waiting time compromises the functionality of health system, yet there is dearth of research on the factors contributing to long waiting times

and interventions implemented to reduce waiting times within PHC facilities in South African context. The results of this study will not only be an important contribution to the body of evidence on waiting times and related interventions but may be crucial for designing healthcare delivery and patient satisfaction interventions in South Africa and similar settings.

## Methods

### Scoping review

We conducted a scoping review study. Due to its ability to facilitate the mapping of new concepts, and types of evidence and identifies related gaps [25]. This review was guided by the Arksey and O'Malley methodological framework [26]. The framework involves (i) identifying the research question, (ii) identifying relevant studies, (iii) study selection, (iv) charting the data, and (v) collating, summarising, and reporting results. The review protocol has been registered with the Open Science Framework database (registration number: osf.io/fpzwu). We used the Preferred Reporting Items for Systematic Reviews and Meta-Analyses extension for Scoping Reviews (PRISMA-ScR) checklist to ensure each step in the scoping review process was standardized [27] (S1 Appendix).

### Ethical approvals

This study was approved by the UKZN Biomedical Research Ethics Committee (BREC) approval number BREC/00003637/2021.

### Identifying the research question

The general research question was: "What is known from the existing literature on the factors contributing to long waiting times and interventions to reduce waiting times within PHC facilities in South Africa?"

Sub-questions were:

- Which factors contribute to long waiting times within PHC facilities in South Africa?

- Which interventions have been implemented to reduce waiting times within PHC facilities in South Africa?

### Eligibility criteria

- The Population-Concept-Context (PCC) framework [28] was used to determine the eligibility of the research question as illustrated below (Table 1).

### Inclusion and exclusion criteria

We included original research articles reporting information regarding factors contributing to long waiting times and interventions to reduce waiting times within PHC facilities in South

**Table 1. PCC framework for determining the eligibility of the research question.**

| Criteria | Determinants |
|---|---|
| Population | Patients and healthcare workers of all ages |
| Concept | Articles focusing on the factors contributing to long waiting times and the interventions implemented to reduce waiting times within PHC facilities |
| Context | Primary healthcare settings in South Africa |

Africa. Articles addressing the research question and utilising any study design published in peer-reviewed journals and in grey literature (research or information material that is not widely published or available and usually does not go through the peer-review process including a wide range of documents and sources, such as reports, theses, conference proceedings, working papers, government documents, non-profit organization publications, patents, and unpublished research studies), were sourced. Articles written in English and published from January 2010 until October 2022 were considered. We excluded case studies, commentaries, editorials, and correspondences. Non-English studies were excluded as previous studies showed that it had a minimal effect on the overall conclusions [29, 30].

## Literature search

Search terms used were waiting times, outpatient department, factors, interventions, primary healthcare facilities, and South Africa, through PubMed/MEDLINE, Google Scholar, Science Direct, World Health Organization (WHO) library databases, and grey literature. The search strategy was piloted to check the appropriateness of selected electronic databases and keywords. Consistency across databases was maintained where possible. Boolean terms (AND, OR) were used to separate the keywords during the search. Medical Subject Headings (MeSH) terms were also included in the search (S2 Appendix).

## Study selection

We conducted a comprehensive title screening; all studies that did not address the study's research question were excluded, along with all the duplicates. All included studies for abstract screening were uploaded to Endnote X7 software. UKZN library services were sought for articles that were difficult to find, failing which we contacted authors directly to request full copies of the articles. The final Endnote database was shared among the review team for abstract screening, i.e., two independent reviewers screened the abstracts and full articles guided by the inclusion criteria. Any discrepancies in the results of the abstract screening were resolved through discussion until a consensus was reached. A third screener was consulted to resolve discrepancies in full-article screening results.

## Charting the data

A standardized data extraction tool (Google Form) was used to extract data from all the variables that focused on answering the research questions, namely: author and date, study population, study design, study setting, factors, and interventions. The data extraction tool was piloted by all the reviewers to confirm consistency. The data was extracted by the first author (UIN) and underwent assessment by another reviewer (KWH) independently.

## Collating, summarising and reporting the results

This study mapped the existing evidence and summarised the findings as presented across articles. A narrative account of the data extracted from the included studies was analysed using thematic analysis, which is a method that is often used to analyse data in primary qualitative research [31]. We used NVIVO® software version 10 to code the data from the included studies. Emerging themes were identified, and data was coded according to these themes. Data was extracted around the following outcomes: factors contributing to long waiting times and interventions to reduce waiting times. Where the presented data was ambiguous or missing, the corresponding authors were contacted for clarification on the process of data extraction.

## Results

Our search yielded 1021 articles, but after title screening and elimination of duplicates, two hundred and fifty (250) eligible studies were retained. Two hundred (200) articles were excluded after abstract screening, thus reducing the number of articles eligible for full article screening to fifty (50) articles. After the full article screening, forty-one (41) articles were excluded, and the remaining nine (9) articles were included for data extraction. Results of the articles screened are presented in (Fig 1).

### Characteristics of the included studies

All the included studies were conducted in PHC facilities in South Africa (comprising primary healthcare centres and community healthcare centres (CHCs). Seven papers identified the factors contributing to waiting time [33–39], and five papers mentioned effective interventions implemented in the reduction of waiting times within PHC facilities [33, 34, 39–41]. The characteristics of the included studies are illustrated in (Table 2). The study findings have been classified into three main themes: patient factors, staff factors, and administrative systems.

## Factors contributing to long waiting time in PHC facilities

### Patient factors

Access to the facility, through public transport, was a major factor that affected the proper implementation of the appointment scheduling system, as some of the surveyed facilities were in locations with poor access to public transport. This resulted in patients always arriving very early in the morning before the clinic opening time, regardless of their appointment time slot, thereby causing a long queue [34–36, 38]. The fluctuating patient numbers was another important factor that influenced waiting time as some days witnessed higher patient volume than others and the pattern was less predictable [37]. Patients' lack of adherence to the booking system at the CHC resulted in long waiting times and patient overload [38]. A study showed that the possible reason for patients' waiting times was because some patients had to see a physician and a professional nurse on the same day and so had to book physicians' appointments for specific days thereby causing delays [33].

### Staff factors

A before and after study reported that factors associated with long waiting times were due to high work volume [36, 39], there was also a lack of efficiency whereby staff members at the service points were busy with something else other than attending to patients while they were waiting, a mismatch where patients are available to be attended to but staff members were not at the service point, queuing problems where patients do not queue in the correct order and staff are not attending to patients in the order that they arrive at the service point, and inappropriate high service time [36]. A study showed that a shortage of staff contributed to long waiting times and the staff on duty arrived late and did not start working as soon as they arrived [37, 39]. Staff meetings during clinic operating hours also influenced waiting times [37]. Poor time management was another factor, whereby the staff arrived at work late, had unauthorized extensions of meal breaks, preoccupied with non-work related matters and departed before the work closing hour [38]. A study reported that communication was another factor that influenced waiting time; hence, few of the patients were foreign nationals and did not understand the local language(s). The lack of understanding and poor communication resulted in the women feeling lost and confused, thereby causing a delay in their antenatal care activities [37, 39].

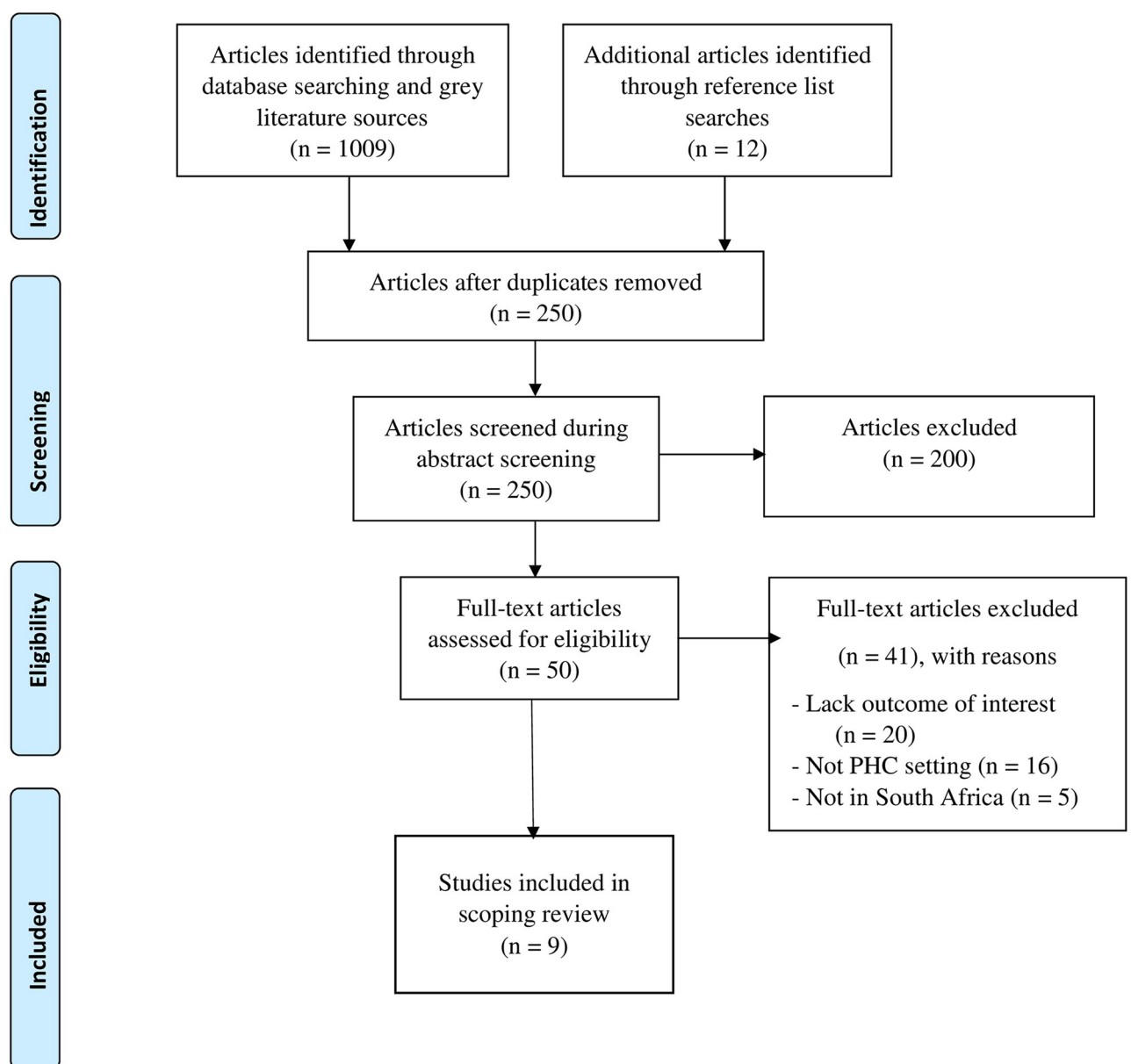

**Fig 1. The Preferred Report Items for Systematic and Meta-Analysis (PRISMA) flow chart for the selection and screening of studies done in this research [32].**

Staff attitude and behaviours were seen to contribute to extended clinic visits as the staff worked at a slow pace and had conversations with their colleagues during consultations [37, 39]. Patients reported that unfavourable staff attitudes discouraged them from visiting the clinic, and poor impressions of staff attitudes impacted assessments of services [39].

## Administrative systems

The use of an appointment scheduling system, such as diaries and notebooks, to schedule clinic appointments was ineffective, as patients were often overbooked because of the limited

**Table 2. Characteristics of the 10 included studies.**

| Author, year | Study design | Study setting | Study Population | Factors | Interventions |
|---|---|---|---|---|---|
| Anna-Therese Swart et al. 2018 [33] | A quantitative, exploratory, typical descriptive pre-test—post-test design | PHC facilities in a sub-district of the Northwest province of South Africa | The target patient populations in this study comprised 333 PHC patients in phase 1 for collecting the baseline data obtained during week one and 332 PHC patients during phase 2, which was the second week after the CTS intervention had been implemented. | A possible reason for patients' waiting times was that some patients had to see a physician and a professional nurse. | The Cape Triage Score (CTS) system showed practical significance where triage was applied, and referral was much quicker to the correct health professional. |
| B. A. Egbujie et al. 2018 [34] | Before-and-After operational research | 10 facilities in the three subdistricts of Amajuba District. Nine of the 10 selected facilities are in rural or periurban areas and one is in an urban area. | A total of 1,763 patients from nine facilities were successfully surveyed: 860 in the initial survey and 903 in the follow-up survey. | Overbooking of patient appointment Many patients arriving early in the morning. | Observed changes were positive when a clinic appointment system was successfully implemented and negative when this was unsuccessful. |
| Brendan Maughan-Brown et al. 2018 [35] | A pilot randomised controlled trial (RCT) | Mobile clinic in Cape Town, South Africa | Men and women 18 years and older (N = 41) | Delays due to lost clinic folders and test results, difficulty navigating the clinic system, and a large number of patients arriving early in the morning. | N/A |
| Johann Daniels et al. 2017 [36] | A before and after study | Several clinics in Cape Town, South Africa | All patients arriving at the clinic on one specific day were included in the sample. All clinic managers were included in the sample. | High workloads; large batches of patients arriving at a clinic before the clinic opening time. Logistical problems such as bottlenecks in patient flow, queuing problems, and inappropriately long service times. | N/A |
| Justine C. Baron and Doreen Kaura 2021 [37] | A qualitative methodology with a single case study design | This study was conducted in an antenatal clinic within a Midwife Obstetric Unit (MOU), in Western Cape, South Africa. | The study sample consisted of 14 participants, of which 12 were pregnant women aged between 18 and 46 years old. | Shortage of staff. | N/A |
| Solly Ratsietsi Makua and Sisinyana Hannah Khunou 2022 [38] | Qualitative exploratory descriptive and contextual design | Community Health Centers (CHCs) in Gauteng Province, South Africa | Eight nurse managers. | Records and patient administration systems deficiencies; poor time management and lack of adherence to booking system by patients. | N/A |
| Dudu G. Sokhela et al. 2013 [39] | A descriptive qualitative survey | Primary health care facilities in eThekwini district, South Africa | 83 health care users | Increased workload, Health care provider attitudes, Shortage of staff. | The fast queue assisted in directing users to the respective queues, reducing waiting time, and keeping users satisfied with the use of signposts where there was a lack of human resources. |
| Bhakti Hansoti et al. 2016 [40] | A pre/post-evaluation study design | Primary healthcare clinics in Cape Town, South Africa | A total of 3064 children were enrolled in the study. | N/A | The implementation of SCREEN reduced waiting times for all critically ill children and had a positive impact on the left without being seen (LWBS) rates in all clinics. |

*(Continued)*

**Table 2.** (Continued)

| Author, year | Study design | Study setting | Study Population | Factors | Interventions |
|---|---|---|---|---|---|
| Bhakti Hansoti et al. 2017 [41] | A prospective, observational implementation-effectiveness hybrid study | Several PHCs in Cape Town, South Africa | In Phase I, 1600 (92.38%) of 1732 children presenting to 4 clinics, had sufficient data for analysis and comprised the control sample. In Phase II, all 3383 of the children presenting to the 26 clinics during the sampling time frame had sufficient data for analysis. | N/A | The Sick Children Require Emergency Evaluation Now (SCREEN) programme effectively reduced waiting times for critically ill children in the primary healthcare setting |

pages on the notebooks and diaries. This results in overbooking as the facilities are not able to determine upfront how many patients they have already scheduled for a particular day [34]. Health service challenges included difficulty in securing appointments, administrative mistakes (especially lost clinic folders and test results), difficulty navigating the clinic system (e.g., failure to collect a queue card or waiting for incorrect services), and negative clinic-patient interactions [35]. A logistical problem due to the lack of equipment or available rooms to attend to patients, flow problems where staff are available to see patients, and the patients get delayed at some other service point [36, 37]. The biggest contributor to waiting time was that follow-up visits were seen before the women who arrived for their initial visits [37]. Baron and Kaura [37], stated that patient flow increased long waiting times, and this is a result of a poor queuing system, a lack of signage, and the ineffective orientation of patients within healthcare facilities. The unavailability of equipment negatively impacted on waiting time [37]. The inefficiencies in the records and patient administration systems contributed to long waiting times in the PHC facilities [38].

## Interventions to reduce waiting times in PHC facilities

Five studies included in this scoping review showed the interventions implemented in PHC facilities to reduce waiting time [33, 34, 39–41]. The interventions have been classified into two themes: Manual-based waiting time reduction systems and electronic-based waiting time reduction systems.

### Manual-based waiting time reduction system

A study showed that the implementation of the Cape Triage Score (CTS) system in emergency departments in the Northwest province of South Africa reduced waiting times, and referral was much quicker to the correct health professionals and the hospitals [33]. The patients were triaged by professional nurses with different codes based on the outcomes of their vital signs or the condition of the patient. Stickers were placed on each patient's hand and file. Patients with a red colour code were sent to professional nurses and physicians immediately, and those with an orange colour code were seen by professional nurses within 10 minutes. Patients with yellow colour codes were seen within 60 min, and patients with green colour codes had to be seen within 240 minutes. The blue colour code pertained to patients who had passed away and needed certification [33].

A qualitative study in the eThekwini district showed that the Fast Queue strategy was also effective in addressing the overcrowding problem in a health facility among healthcare users [39]. The users reported that they were attended to quickly as the fast queue enabled them to

be separated into different queues, and mothers of young babies liked the fact that they were seated away from the sick people, thus protecting their children from possible infections [39].

### Electronic-based waiting time reduction system

A positive change in waiting time reduction was observed when a patient appointment scheduling system was successfully implemented [34]. Facilities that improved their patient waiting time recorded a wider and more evenly spread patient arrival pattern after the intervention compared with before [34].

The implementation of the Sick Children Require Emergency Evaluation Now (SCREEN) tool on the waiting times of critically ill children reduced waiting times for all critically ill children and had a positive impact on the left without being seen (LWBS) rates in all clinics [40]. A quick response (QR) code was placed on the child's clothing before they entered the clinic and joined the queue. A QR code is a machine-readable code consisting of an array of black and white squares. The nurses were asked to scan the QR codes to record the Integrated Management of Childhood Illnesses (IMCI) category as (red/yellow/green) to delineate each child's severity of illness [40].

A study showed that a simple screening tool implemented for use by laypersons significantly reduced waiting time for critically ill children in primary healthcare clinics [41].

### Discussion

Our study identified and classified three major factors contributing to waiting time as patient factors, staff factors, and administrative systems, and the interventions that were implemented successfully in reducing waiting times within PHCs were grouped into two main themes: manual-based waiting time reduction systems and electronic-based waiting time reduction systems. The included studies showed that most patients usually arrive and queue outside the clinics very early before the clinic doors or gates are opened [34–36, 38]. In South Africa, it's typical to arrive early and wait in queue outside of clinics. This problem is attributable to the lack of an electronic appointment scheduling system [42], or poor access to the facility using public transport, given that the majority of the users of public healthcare facilities rely on public transport, which often operates at rigid times, such as the early mornings and afternoons in most semi-urban (townships) and rural areas of South Africa [34]. This also poses security risks, in addition to discomforts that may be imposed by inclement weather. A study by Egbujie et al. [34] showed that some facilities recorded an evenly spread patient arrival pattern through proper appointment scheduling, and this reduced the daily overbooking of patients and ensured an even spread of booked patients throughout the day [34]. Patients not honouring their appointments as scheduled impacted adversely on patients' waiting time [38]. A before and after study by Daniels et al. [36], stated that scheduling appointments for quieter times and days in the week and encouraging patients to come at less busy times in the day also reduced waiting times. There is, therefore, a need for urgent scale-up of electronic appointment scheduling systems in all PHCs in South Africa, as it will significantly reduce patient waiting times.

Lack of signage and a poor queuing system reduced patient flow and increased waiting time, as previously reported [37]. Staff attitudes and behaviours were seen to contribute to extended clinic visits, as some patients stated that negative staff attitudes deterred them from visiting the clinic [37, 39]. Continued in-service training for healthcare workers will help improve effective communication and patient satisfaction. Staff reporting late for work, having unauthorized meal breaks, preoccupied with non-work-related matters, and leaving work

ahead of time [36, 38]. Healthcare workers need to prioritise attending to patients as soon as they arrive, report to work timeously, and close at the appropriate time.

The language barrier was reported to be a factor causing a delay, as few patients were foreign nationals and did not understand the health workers. The lack of understanding and poor communication resulted in the women feeling lost and confused [37], hence, healthcare workers should be accommodating and use the English language for foreign nationals.

Our study identified some interventions that were successfully implemented in reducing waiting times within primary healthcare facilities, and they were grouped into two main themes: manual-based waiting time reduction systems and electronic-based waiting time reduction systems. The manual-based intervention system showed a significant reduction in waiting time, as triage was applied and professional nurses were able to prioritise attending to more seriously ill patients, resulting in quicker referral to the correct health professionals [33]. Furthermore, the use of fast queues and queue marshals helped direct users to their respective queues and reduced waiting time [39]. The use of multiple, managed queues was generally well-received by attendees, according to a qualitative study on the use of a "Fast Queue" in KZN clinics. This was especially true if there was smooth (i.e., unidirectional) flow and effective communication with health workers [39].

The electronic-based waiting time reduction system showed a positive change in waiting time reduction when the patient appointment scheduling system was properly implemented [34]. Facilities that improved their patient waiting times recorded a wider and more evenly spread patient arrival pattern after the intervention [34]. The SCREEN program enabled the quick identification of critically ill children and expedited their care at the point of entry into a clinic, and this significantly reduced waiting times for critically ill children in PHC clinics [40, 41].

PHC and community health center (CHC) facility managers may need to constantly strategize on measures to reduce patient waiting times, educate the staff on the best way to approach patients, to increase patient satisfaction and quality of care.

## Strengths and limitations

To our knowledge, this is the first scoping review to map evidence on the factors contributing to long waiting times and interventions that have been implemented successfully in reducing waiting times within primary healthcare facilities in South Africa. A few studies were included after a thorough systematic search strategy was conducted. There is still, however, the possibility that relevant articles were omitted. Nevertheless, the absence of some relevant articles does not significantly impact the findings of this study. Future studies may be directed at other jurisdictions and developing countries in Africa and elsewhere, as well as studies designed to evaluate the impact of measures designed to reduce patient waiting at public healthcare facilities.

## Conclusions

There was a paucity of studies focusing on factors contributing to long waiting times and interventions implemented to reduce waiting time in primary healthcare facilities in South Africa. Our study findings show the need for future primary research focusing on implementing the use of an electronic appointment scheduling system and database within the PHC facilities for effective patient administrative records, as patients faced difficulties in securing appointments and administrative mistakes occurred due to lost clinic folders and test results. There is a need for continuous training to improve staff attitudes and behaviours as it will positively reduce waiting time, and improve quality of care, and patient satisfaction.

## Supporting information

**S1 Appendix. Preferred reporting items for systematic reviews and meta-analyses extension for scoping reviews (PRISMA-ScR) checklist.**
(DOCX)

**S2 Appendix. PubMed search strategy.**
(DOCX)

## Author Contributions

**Conceptualization:** Ugochinyere I. Nwagbara, Khumbulani W. Hlongwana, Sylvester C. Chima.

**Data curation:** Ugochinyere I. Nwagbara, Khumbulani W. Hlongwana, Sylvester C. Chima.

**Formal analysis:** Ugochinyere I. Nwagbara, Sylvester C. Chima.

**Investigation:** Ugochinyere I. Nwagbara, Khumbulani W. Hlongwana, Sylvester C. Chima.

**Methodology:** Ugochinyere I. Nwagbara, Khumbulani W. Hlongwana, Sylvester C. Chima.

**Supervision:** Khumbulani W. Hlongwana, Sylvester C. Chima.

**Validation:** Khumbulani W. Hlongwana, Sylvester C. Chima.

**Writing – original draft:** Ugochinyere I. Nwagbara, Khumbulani W. Hlongwana, Sylvester C. Chima.

**Writing – review & editing:** Ugochinyere I. Nwagbara, Khumbulani W. Hlongwana, Sylvester C. Chima.

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
