## [Decision Letter · Decision Letter 0]

2 Oct 2023

PONE-D-23-01980Mapping evidence on the factors contributing to long waiting times and interventions to reduce waiting times within primary health care facilities in South Africa: a scoping review.PLOS ONE Dear Dr. Nwagbara Thank you for submitting your manuscript to PLOS ONE. After careful consideration, we feel that it has merit but does not fully meet PLOS ONE’s publication criteria as it currently stands. Therefore, we invite you to submit a revised ver\\sion of the manuscript that addresses the points raised during the review process. Please submit your revised manuscript by 10 October 2023. If you will need more time than this to complete your revisions, please reply to this message or contact the journal office at plosone@plos.org. Please include the following items when submitting your revised manuscript:A rebuttal letter that responds to each point raised by the academic editor and reviewer(s). You should upload this letter as a separate file labeled 'Response to Reviewers'.A marked-up copy of your manuscript that highlights changes made to the original version. You should upload this as a separate file labeled 'Revised Manuscript with Track Changes'.An unmarked version of your revised paper without tracked changes. You should upload this as a separate file labeled 'Manuscript'.If applicable, we recommend that you deposit your laboratory protocols in protocols.io to enhance the reproducibility of your results. Protocols.io assigns your protocol its own identifier (DOI) so that it can be cited independently in the future. For instructions see: https://journals.plos.org/plosone/s/submission-guidelines#loc-laboratory-protocols. Additionally, PLOS ONE offers an option for publishing peer-reviewed Lab Protocol articles, which describe protocols hosted on protocols.io. Read more information on sharing protocols at https://plos.org/protocols?utm_medium=editorial-email&utm_source=authorletters&utm_campaign=protocols. We look forward to receiving your revised manuscript. Kind regards, Arun Kumar SharmaAcademic EditorPLOS ONE Journal Requirements:

Reviewers' comments: Reviewer's Responses to Questions Comments to the Author 1. Is the manuscript technically sound, and do the data support the conclusions? The manuscript must describe a technically sound piece of scientific research with data that supports the conclusions. Experiments must have been conducted rigorously, with appropriate controls, replication, and sample sizes. The conclusions must be drawn appropriately based on the data presented. Reviewer #1: Partly Reviewer #2: Yes 2. Has the statistical analysis been performed appropriately and rigorously? Reviewer #1: N/A Reviewer #2: N/A 3. Have the authors made all data underlying the findings in their manuscript fully available? The PLOS Data policy requires authors to make all data underlying the findings described in their manuscript fully available without restriction, with rare exception (please refer to the Data Availability Statement in the manuscript PDF file). The data should be provided as part of the manuscript or its supporting information, or deposited to a public repository. For example, in addition to summary statistics, the data points behind means, medians and variance measures should be available. If there are restrictions on publicly sharing data—e.g. participant privacy or use of data from a third party—those must be specified. Reviewer #1: Yes Reviewer #2: Yes 4. Is the manuscript presented in an intelligible fashion and written in standard English? PLOS ONE does not copyedit accepted manuscripts, so the language in submitted articles must be clear, correct, and unambiguous. Any typographical or grammatical errors should be corrected at revision, so please note any specific errors here. Reviewer #1: No Reviewer #2: Yes 5. Review Comments to the Author Please use the space provided to explain your answers to the questions above. You may also include additional comments for the author, including concerns about dual publication, research ethics, or publication ethics. (Please upload your review as an attachment if it exceeds 20,000 characters) Reviewer #1: Dear authors, I enjoyed reading about your study which I find extremely important. Despite the importance of the study to increase patient centeredness, I have some comments which need to be addressed:-I strongly recommend to rewrite the introduction which needs to be build more logically. Now, you go from general information to more specific and back. You also often repeat the same information.-Also, I would suggest to use a proofreading service because some parts are not written well (not clear enough, grammar, enumeration instead of separate sentences, etc.)-You should also emphasize the novelty of this study.-RQ: the general RQ should be: “What is known about…?” and the specific RQ should be: “Which factors contribute to an increase in the waiting time? …-Arksey & O’Malley: normally there are 6 steps. You did not consult the stakeholders. Why not? In this case it seems very interesting to ask them what they think about these results.-Repetition in inclusion section and identifying section!-What do you mean with “grey literature”. You should at least describe what your definition is and where you have looked for these papers.-Discussion: just a summary of the results above, and no comparison with the existing literature. Was this found elsewhere??? Reviewer #2: It is a well written manuscript, presenting scoping review on an important issue fundamental to ensuring health for all. The PRISMA record screening flow chart makes it clear that there was a paucity of evidence on the topic, justifying the study. Following are few minor revisions and suggestions for the authors:1. Abstract:a. Word count is above 300 words. Please shorten.b. Objective (This scoping review explored and mapped literature on the factors contributing to long waiting times and interventions to reduce waiting times within PHC facilities in South Africa) is mentioned in Methods. Shift to Introduction.c. Conclusion: Mentions "Therefore, implementing the use of an electronic database within the PHC facilities for effective patient administrative records will enable patients to successfully secure appointments" - Use of only database will not solve the scheduling problem. Thus, the sentence should be rephrased to "implementing the use of an electronic appointment scheduling system and database". This comment holds true for the main text of the manuscript as well. 2. Resultsa. Page 16, Lines 11-13: "A study reported that communication was another factor that influenced waiting time, hence few of the participants were foreign nationals and did not understand local language/s." - Not clear who the "participant" is here. From the conclusion, it seems the authors wished to write "patients". Rephrase as appropriate. 3. Conclusionsa. Page 20, Lines 5-6: "The included studies showed that most patients were not satisfied with the amount of time spent in healthcare facilities while waiting for medical assistance." This is new information in Discussion, no corresponding results presented to arrive at this conclusion.b. Page 20, Lines 23-24: "As a result, some patients would prefer to miss their appointments and go to other clinics where the staff attitude is more favourable and this in turn creates a disorganised patient flow and increased waiting times (40, 42)." his is new information in Discussion, no corresponding results presented to arrive at this conclusion.c. Page 21, Lines 3-4: "Staff reporting late for work, having unauthorized meal-breaks, preoccupied with non-work related matters and leaving work ahead of time (39, 41)." This is new information in Discussion, no corresponding results presented to arrive at this conclusion.  Overall, the authors are suggested to include these new statements pertaining to the effects of long waiting times in the results section (and not discussion section for the first time) wherever appropriate. 6. PLOS authors have the option to publish the peer review history of their article (what does this mean?). If published, this will include your full peer review and any attached files.  Do you want your identity to be public for this peer review? For information about this choice, including consent withdrawal, please see our Privacy Policy. Reviewer #1: No Reviewer #2: Yes: Dr Shaileja Yadav    While revising your submission, please upload your figure files to the Preflight Analysis and Conversion Engine (PACE) digital diagnostic tool, https://pacev2.apexcovantage.com/. PACE helps ensure that figures meet PLOS requirements. To use PACE, you must first register as a user. Registration is free. Then, login and navigate to the UPLOAD tab, where you will find detailed instructions on how to use the tool. If you encounter any issues or have any questions when using PACE, please email PLOS at figures@plos.org. Please note that Supporting Information files do not need this step.

---

## [Author Response · Author response to Decision Letter 0]

29 Dec 2023

Author’s response to the reviews.

Manuscript title: Mapping evidence on the factors contributing to long waiting times and interventions to reduce waiting times within primary health care facilities in South Africa: a scoping review

Manuscript PONE-D-23-01980

We are very grateful for the reviews provided by the editor and the reviewers of this manuscript. The comments were very useful. We have revised the manuscript according to the reviewer’s comments. Below is the point-by-point response to the reviewer’s comments. Please find attached a revised version of the manuscript with tracked changes highlighted in ‘red’.

# Editor’s comment Authors’ responses

 Page and Line Number/s 

1 I strongly recommend to rewrite the introduction which needs to be build more logically. Now, you go from general information to more specific and back. You also often repeat the same information Thanks for the recommendation, we have re-written the introduction. Page 4, lines 74-110

2 Also, I would suggest to use a proofreading service because some parts are not written well (not clear enough, grammar, enumeration instead of separate sentences, etc.) Thanks for the suggestion, we have profred our manuscript Whole manuscript

3 You should also emphasize the novelty of this study Thanks, this has been emphasized Page 5, lines 105-110

4 RQ: the general RQ should be: “What is known about…?” and the specific RQ should be: “Which factors contribute to an increase in the waiting time? … Thanks, this has been corrected Page 6, lines 135-141

5 Arksey & O’Malley: normally there are 6 steps. You did not consult the stakeholders. Why not? In this case it seems very interesting to ask them what they think about these results. The consultation stage is optional in Arksey and O’Malley’s original framework for conducting scoping reviews. However, in this scoping review, this stage will be omitted as the focus is mapping evidence on the factors contributing to long waiting times and interventions to reduce waiting times within primary health care facilities in South Africa, so there will not be any consultation of stakeholders as the stakeholders will be consulted during the qualitative phase of the research. Page 6

6 Repetition in inclusion section and identifying section! Thanks, this has been corrected Page 7-8

7 What do you mean with “grey literature”. You should at least describe what your definition is and where you have looked for these papers Thanks, we have clarified our definition of grey literature. Page 7, lines 153-157

8 Discussion: just a summary of the results above, and no comparison with the existing literature. Was this found elsewhere??? We have updated our discussion section Pages 19-21

Reviewer 2 

1 Abstract: 

A Word count is above 300 words. Please shorten Thanks, the abstract section has been shortened Pages 2-3, lines 28-54

B Objective (This scoping review explored and mapped literature on the factors contributing to long waiting times and interventions to reduce waiting times within PHC facilities in South Africa) is mentioned in Methods. Shift to Introduction. We have removed the sentence from the methods section 

C Conclusion: Mentions "Therefore, implementing the use of an electronic database within the PHC facilities for effective patient administrative records will enable patients to successfully secure appointments" - Use of only database will not solve the scheduling problem. Thus, the sentence should be rephrased to "implementing the use of an electronic appointment scheduling system and database". This comment holds true for the main text of the manuscript as well. Thanks for the observation, we have corrected the sentence as suggested Pages 2-3, line 53 and Page 22, lines 175-179

2 Results 

a Page 16, Lines 11-13: "A study reported that communication was another factor that influenced waiting time, hence few of the participants were foreign nationals and did not understand local language/s." - Not clear who the "participant" is here. From the conclusion, it seems the authors wished to write "patients". Rephrase as appropriate.

 Thanks, we have rephrased as appropriate Page 16, lines 39-41

3 Conclusions 

A Page 20, Lines 5-6: "The included studies showed that most patients were not satisfied with the amount of time spent in healthcare facilities while waiting for medical assistance." This is new information in Discussion, no corresponding results presented to arrive at this conclusion. Thanks, the sentence has been corrected Page 19, lines 112-113

B Page 20, Lines 23-24: "As a result, some patients would prefer to miss their appointments and go to other clinics where the staff attitude is more favourable and this in turn creates a disorganised patient flow and increased waiting times (40, 42)." his is new information in Discussion, no corresponding results presented to arrive at this conclusion Thanks, the sentence has been corrected Page 16, lines 46-47 and Page 20, lines 128-130

C Page 21, Lines 3-4: "Staff reporting late for work, having unauthorized meal-breaks, preoccupied with non-work related matters and leaving work ahead of time (39, 41)." This is new information in Discussion, no corresponding results presented to arrive at this conclusion. Thanks, the sentence has been corrected Page 16, lines 37-39, page 20, lines 131-133

 Overall, the authors are suggested to include these new statements pertaining to the effects of long waiting times in the results section (and not discussion section for the first time) wherever appropriate. Thanks, we have included the statements in the result section

---

## [Decision Letter · Decision Letter 1]

7 Feb 2024

Mapping evidence on the factors contributing to long waiting times and interventions to reduce waiting times within primary health care facilities in South Africa: a scoping review.

PONE-D-23-01980R1

Dear Dr. Nwagbara,

We’re pleased to inform you that your manuscript has been judged scientifically suitable for publication and will be formally accepted for publication once it meets all outstanding technical requirements.

Kind regards,

Arun Kumar Sharma

Academic Editor

PLOS ONE

Additional Editor Comments (optional):

Reviewers' comments:

Reviewer's Responses to Questions

**Comments to the Author**

1. If the authors have adequately addressed your comments raised in a previous round of review and you feel that this manuscript is now acceptable for publication, you may indicate that here to bypass the “Comments to the Author” section, enter your conflict of interest statement in the “Confidential to Editor” section, and submit your "Accept" recommendation.

Reviewer #2: All comments have been addressed

2. Is the manuscript technically sound, and do the data support the conclusions?

Reviewer #2: Yes

3. Has the statistical analysis been performed appropriately and rigorously? 

Reviewer #2: N/A

4. Have the authors made all data underlying the findings in their manuscript fully available?

Reviewer #2: Yes

5. Is the manuscript presented in an intelligible fashion and written in standard English?

Reviewer #2: Yes

6. Review Comments to the Author

Reviewer #2: Comment 2a. (Page 16, lines 39-41) “A study reported that communication was another factor that influenced waiting time; hence, few of the patients were foreign nationals and did not understand the local language(s)” – Change to “hence” to “since” for grammatical accuracy.

7. PLOS authors have the option to publish the peer review history of their article (what does this mean?). If published, this will include your full peer review and any attached files.

Reviewer #2: **Yes: **Dr Shaileja Yadav

---

## [Editor Report · Acceptance letter]

9 Jul 2024

PONE-D-23-01980R1 

PLOS ONE

Dear Dr. Nwagbara, 

I'm pleased to inform you that your manuscript has been deemed suitable for publication in PLOS ONE. Congratulations! Your manuscript is now being handed over to our production team.

Kind regards, 

on behalf of

Professor Arun Kumar Sharma 

Academic Editor

PLOS ONE